# Antiviral Effects of 5-Aminolevulinic Acid Phosphate against Classical Swine Fever Virus: In Vitro and In Vivo Evaluation

**DOI:** 10.3390/pathogens11020164

**Published:** 2022-01-27

**Authors:** Shizuka Hirose, Norikazu Isoda, Loc Tan Huynh, Taksoo Kim, Keiichiro Yoshimoto, Tohru Tanaka, Kenjiro Inui, Takahiro Hiono, Yoshihiro Sakoda

**Affiliations:** 1Laboratory of Microbiology, Department of Disease Control, Faculty of Veterinary Medicine, Hokkaido University, Kita 18, Nishi 9, Kita-ku, Sapporo 060-0818, Hokkaido, Japan; shizuka-1612asteroid@eis.hokudai.ac.jp (S.H.); nisoda@vetmed.hokudai.ac.jp (N.I.); huynhtanloc@vetmed.hokudai.ac.jp (L.T.H.); ts-kim@vetmed.hokudai.ac.jp (T.K.); hiono@vetmed.hokudai.ac.jp (T.H.); 2International Institute for Zoonosis Control, Hokkaido University, Kita 20, Nishi 10, Kita-ku, Sapporo 001-0020, Hokkaido, Japan; 3Department of Veterinary Medicine, College of Agriculture, Can Tho University, Campus II, 3/2 Street, Ninh Kieu, Can Tho 900000, Vietnam; 4Neopharma Japan Co., Ltd., Koujimachi 6-2-6, Chiyoda-ku, Tokyo 102-0083, Japan; keiichiro.yoshimoto@neopharmajp.com (K.Y.); o.tohru.tanaka@neopharmajp.com (T.T.); 5National Center for Veterinary Diagnosis, Tan Chung Chua, Hien Ninh, Soc Son, Hanoi 122000, Vietnam; ken.inui@gmail.com

**Keywords:** 5-aminolevulinic acid phosphate, antiviral effect, classical swine fever virus, protoporphyrin IX

## Abstract

The inhibitory effects of 5-aminolevulinic acid phosphate (5-ALA), an important amino acid for energy production in the host, against viral infections were previously reported. Here, the antiviral effects of 5-ALA against classical swine fever virus (CSFV) belonging to the genus *Pestivirus* in the *Flaviviridae* family and its possible mechanisms were investigated. CSFV replication was suppressed in swine cells supplemented with 5-ALA or its metabolite, protoporphyrin IX (PPIX). The infectivity titer of CSFV was decreased after mixing with PPIX extracellularly. In addition, the activities of the replication cycle were decreased in the presence of PPIX based on the CSFV replicon assay. These results showed that PPIX exerted antiviral effects by inactivating virus particles and inhibiting the replication cycle. To evaluate the in vivo efficacy of 5-ALA, pigs were supplemented daily with 5-ALA for 1 week before virus inoculation and then inoculated with a virulent CSFV strain at the 10^7.0^ 50% tissue culture infectious dose. The clinical scores of the supplemented group were significantly lower than those of the nonsupplemented group, whereas the virus growth was not. Taken together, 5-ALA showed antiviral effects against CSFV in vitro, and PPIX played a key role by inactivating virus particles extracellularly and inhibiting the replication cycle intracellularly.

## 1. Introduction

Classical swine fever (CSF) is a fatal contagious disease of domestic pigs and wild boars. Due to its huge economic impact, CSF is recognized as one of the most important viral diseases in the pig industry and is listed as a notifiable disease to the World Organization for Animal Health (OIE) [1,2]. According to the official disease status by the OIE in March 2021, only 38 countries have a free status of CSF and 3 countries had a limited CSF-free area, indicating that CSF is still a worldwide threat [3]. CSF is endemic in a large area of Asia, Central and South America, and some Caribbean countries [4]. In these areas, routine vaccination is used to mitigate the impact of CSF outbreaks [2]. As a control strategy recommended by the OIE, slaughtering of affected pigs or prophylactic vaccination for minimizing its impact are mainly implemented instead of treatment. Especially in CSF-free countries where pigs are expected as one of the major export items, rapid containment is applied at the highest priority to meet the requirements for maintaining their official status by the OIE. Therefore, emergency vaccination strategies and slaughtering of suspected and vaccinated pigs are enforced when outbreaks occur. When it is difficult to control the outbreak with previously defined measures, other options, such as adopting antiviral agents, are considered to mitigate the speed of virus spread. For example, the efficacy of T-1105 for foot-and-mouth disease [5] was considered. As a matter of course, antiviral drugs are not applied to infectious diseases, including CSF that causes a great impact on the industry, because infected and suspected animals are slaughtered without medication, and that would be a limitation for the application of antiviral agents. However, especially in countries where CSF is endemic, these drugs are still an option as the first step of disease control. In addition to antiviral agents with a specific effect, supplementation of the compounds with a nonspecific protective effect would be beneficial; for example, vitamin D supplementation is reported to reduce the viral titer and enhance the induction of anti-inflammatory cytokines in mice against influenza virus infection [6]. Thus, supplementation of a nonspecific antiviral compound would be a supportive tool for disease control.

5-Aminolevulinic acid phosphate (5-ALA) is one of the natural amino acids. It is not involved in protein synthesis but is finally metabolized to heme, which composes catalase, cytochrome, or hemoglobin as a cofactor. Externally supplied 5-ALA is imported inside the cell via peptide transporter 1, which is expressed on the cell membrane and mitochondria; similar to the naturally synthesized one; it is converted to coproporphyrinogen III and then protoporphyrin IX (PPIX) and further metabolized to heme in the presence of the ferrous ions provided from the mitochondria or supplemented sodium ferrous citrate (SFC) [7]. SFC provides ferrous ions essential for 5-ALA metabolism into heme caused by ferrochelatase. Heme is further metabolized to biliverdin and carbon monoxide by heme oxygenase-1 (HO-1), and biliverdin is degraded to bilirubin. Without sufficient ferrous ions, PPIX is not further converted but exported outside the cell by an ATP-binding cassette G (ABCG) transporter, whereas a portion of PPIX accumulates in the cell [8]. Recently, in vitro antiviral effects of 5-ALA against feline infectious peritonitis virus and severe acute respiratory syndrome coronavirus 2 (SARS-CoV-2) were reported [9,10], although its antiviral mechanism has not been elucidated yet. Several studies hypothesized that the antiviral effects of metal complexes of protoporphyrin, heme, and HO-1 contribute to the antiviral effects of 5-ALA against several viral pathogens. Metalloporphyrins and heme showed antiviral effects against dengue virus and yellow fever virus, which both belong to the genus *Flavivirus*, family *Flaviviridae*, by inactivating virus particles directly, affecting the adsorption or penetration phase of infection, preventing viral protein synthesis and RNA replication [11], or inhibiting virus entry by interacting with the viral envelope [12].

In this study, the antiviral effects of 5-ALA and its metabolite, PPIX, against CSF virus (CSFV) were found in vitro. Then, it was revealed that PPIX plays a key role in the antiviral mechanisms of 5-ALA. Notably, PPIX showed antiviral effects both inside and outside the cell, possibly by inactivating virus particles extracellularly and inhibiting the replication cycle intracellularly. Finally, the efficacy of 5-ALA against CSFV was evaluated in pigs.

## 2. Results

### 2.1. The Cytotoxicity and Half-Maximal Effective Concentration (EC_50_) of 5-ALA and Its Metabolate

Before investigating the antiviral effects of 5-ALA and its metabolite, the cytotoxicity of 5-ALA, a combination of 5-ALA with SFC, source of ferrous ions, or PPIX, one of the metabolites of 5-ALA, was examined in swine kidney line-L (SK-L) cells (Appendix A). Cell viability in the presence of 5-ALA or 5-ALA with SFC was maintained at ~80%, showing that these compounds did not have critical cytotoxic effects even at high concentrations. Meanwhile, in the case of PPIX, cell viability decreased proportionally at a concentration of >250 µM.

The antiviral effects of 5-ALA and PPIX against CSFV were evaluated by estimating the EC_50_ (Appendix A). With supplementation of 5-ALA only, the EC_50_ value against virulent CSFV vALD-A76 strain was 2.74 mM, whereas, in PPIX supplemented condition, the EC_50_ value was 66.7 µM.

### 2.2. In Vitro Growth Kinetics of CSFV with Supplementation of 5-ALA and Its Metabolite

The viral growth of vALD-A76 at a multiplicity of infection (MOI) of 0.1 with supplementation of the compounds was investigated (Figure 1). Based on the evaluation of EC_50_ against CSFV and other previous studies against coronaviruses [9,10], the final concentrations of 5-ALA and SFC were determined to be 1 and 0.25 mM, respectively. To investigate which step of virus infection 5-ALA and other compounds showed antiviral effects, SK-L cells were supplemented with compounds in pre-treatment, simultaneous, or post-treatment conditions. When 5-ALA was added to the cell simultaneously with virus inoculation, no significant differences were observed in virus growth (Figure 1B). In contrast, pre-treatment with 5-ALA alone significantly suppressed virus growth through the time course of experiment (Figure 1A). In addition, the virus titers in the cells post-supplemented with 5-ALA alone were significantly lower than the nonsupplemented control (Figure 1C). Then, the antiviral effects of 5-ALA supplementation with SFC were investigated. In this condition, most of the 5-ALA was expected to be metabolized into heme, whereas, without SFC, it remained as PPIX rather than heme due to the lack of ferrous ions. Interestingly, virus growth suppression was not observed in pre-supplemented cells at any of the time points, whereas the virus titers in post-supplemented cells were significantly lower compared to the nonsupplemented control (Figure 1A,C).

The observation implied that PPIX plays a pivotal role in the antiviral effects of 5-ALA supplementation against CSFV. In fact, when 5-ALA was added to the cell simultaneously with virus inoculation for 1 h, no antiviral effect was observed regardless of the addition of SFC (Figure 1B), suggesting that PPIX was not produced sufficiently under this condition. Thus, 1 µM of fumitremorgin C (FTC), the inhibitor of the PPIX exporter, was used [8]. The combined supplementation of 5-ALA and FTC results in the intracellular accumulation of PPIX. In the pre-treatment and post-treatment of 5-ALA and FTC, virus growth was dramatically suppressed throughout the experimental period (Figure 1A,C). Consistent with this, with PPIX supplementation, the virus titer remained ≤10^3.0^ TCID_50_/mL in post-treatment, although virus growth suppression was not observed in pre-treatment and simultaneous treatment (Figure 2).

Altogether, 5-ALA and its metabolite PPIX showed an inhibitory effect against virulent CSFV. The results suggested that PPIX plays the main part of the antiviral effect by 5-ALA, especially in the continuous presence of PPIX intracellularly. This compound works not on the virus entry but the other steps of viral growth.

### 2.3. Elucidation of the Action Point by 5-ALA against CSFV vALD-A76 In Vitro

#### 2.3.1. Production and Distribution of PPIX after 5-ALA Supplementation

As the intracellular PPIX concentration was estimated to be important for the antiviral effects of 5-ALA, the PPIX concentration after 5-ALA supplementation was measured. After 5-ALA supplementation in the cell supernatant, the PPIX concentrations inside and outside the cell were measured for 3 days (Figure 3). Intracellular concentration was measured after washing cells thrice with phosphate-buffered saline (PBS), whereas extracellular concentration was measured in the supernatant. With supplementation of 5-ALA alone, the intracellular and extracellular PPIX concentrations increased in a dose- and time-dependent manner (Figure 3A). With the supplementation of the highest concentration (5 mM) of 5-ALA, the intracellular and extracellular PPIX concentrations at 3 days post-supplementation (dps) reached 5 and 50 µM, respectively (Figure 3A) and this gradient was consistent in lower supplementation doses.

Next, the PPIX concentration after 5-ALA supplementation was also measured under the FTC, the inhibitor of the PPIX exporter. In this study, the concentration of FTC was raised to 5 µM since the maximum amount of 5-ALA was 5 mM. When 5 µM FTC was added with 5 mM 5-ALA at 0 dps, the intracellular PPIX concentration reached a maximum of 6 µM on 1 dps and then decreased, whereas the intracellular PPIX concentration did not change in a lower supplementation dose of 5-ALA (Figure 3B). The extracellular concentration increased gradually and reached a maximum at 3 dpi in all three different supplemental concentrations of 5-ALA and FTC, but lower than those without FTC. Collectively, PPIX was produced by 5-ALA supplementation, and most of the produced PPIX was exported extracellularly in natural conditions. Still, some PPIX accumulated in cells after external 5-ALA supplementation.

#### 2.3.2. Direct Inactivation of Virus Particles by PPIX

The PPIX concentration in the extracellular compartment was ~10 times higher than that in the intracellular compartment after 5-ALA supplementation; therefore, the antiviral effects of extracellular PPIX were evaluated. PPIX was mixed with the virus and incubated at 37 °C for 1 h. Then, PPIX was removed from the virus diluent by filtration, and the virus infectivity titer was evaluated. The removal of PPIX was confirmed by fluorescence intensity, the method used to measure PPIX concentrations (Appendix A). As a result, the virus titer of CSFV was suppressed from 5 µM PPIX treatment and it was under the detection limit at a dose of >25 µM (Figure 4), whereas the titer of untreated virus was more than 10^7.0^ TCID_50_/mL. This result demonstrated that PPIX inactivated the virus particles directly.

#### 2.3.3. Inhibition of Replication Cycle of CSFV Replicon by PPIX

Although the intracellular PPIX concentration was far less than the extracellular PPIX, the fact that the combined use of 5-ALA and FTC induced a strong antiviral activity against CSFV motivated the authors to investigate the intracellular antiviral mechanisms of PPIX. For this purpose, luciferase activities of the mono-cistronic replicon (rGPE^–^2GL) were monitored with or without PPIX supplementation (Figure 5). The replication-deficient replicon (rGPE^–^2GL/GAA) without PPIX was used as a control. PPIX was added to the medium soon after the transfection of replicon RNA at the final concentration of 100 µM and incubated for 72 h to measure luciferase activities. The luciferase activity of rGPE^–^2GL was significantly suppressed in the presence of PPIX from 24 h post-transfection until the end of the experiment, whereas that of rGPE^–^2GL/GAA was not (Figure 5). This result showed that intracellular PPIX inhibited the replication cycle of the replicon.

### 2.4. Application of 5-ALA Supplementation in Pigs

As described above, the antiviral effects of 5-ALA were confirmed in vitro; next, the in vivo efficacy of 5-ALA was evaluated in pigs. Nine 2-week-old pigs were divided into the High Dose supplementation group (High Dose; n = 3), Low Dose supplementation group (Low Dose; n = 3), and no-supplementation group (control; n = 3). Pigs in High Dose and Low Dose were fed daily with 5-ALA and SFC at 10 and 15.6 mg/kg and 1 and 1.56 mg/kg, respectively, from 7 days before virus inoculation to 13 dpi.

The average and SD of body temperature of each pig before virus infection (–7 to 0 dpi) were calculated. The development of a high fever was defined as the temperature exceeding more than thrice the SD from their average temperature. Based on this criterion, we evaluated that one pig in High Dose and all pigs in Low Dose and control developed a high fever (Figure 6A). The body temperature of pig #281 decreased severely from 10 dpi, and this pig died at 13 dpi. All pigs showed clinical signs from 3 dpi, and two pigs in Low Dose and all pigs in the control reached >10, whereas the scores were <5 in High Dose (Figure 6B). Significant differences in developing fever were confirmed by the Mann–Whitney U test between High Dose and Low Dose at 10 to 14 dpi and between High Dose and control at 5–8 and 10–12 dpi, respectively (*p* < 0.05).

Regarding virus titers in blood, all three groups followed the same trend; viremia was observed from 3 dpi to the end of the experiment, and virus titers in blood reached ~10^7.0^ TCID_50_/mL in all pigs (Figure 6C; Appendix A). In addition, there was no significant difference in the virus recovery from swab and organ samples (Appendix A). Titers of neutralizing antibodies of all pigs, except for pig #281 that died at 13 dpi, were measured at 14 dpi (Appendix A). Neutralizing antibody titers ranging from 2 to 8 were detected in two pigs in both High Dose and control but were <2 in all pigs in Low Dose.

To confirm the 5-ALA uptake into the body, 5-ALA concentrations in the plasma of the pigs were measured on the first day (–7 dpi) and eighth day (0 dpi; Appendix A). On each day, 5-ALA concentrations in the plasma were detected in a dose-dependent manner, resulting in a significantly higher uptake in High Dose than Low Dose or control on 1 dpi (*p* < 0.05). In addition, the 5-ALA concentration in High Dose on 0 dpi after 7 days supplementation was significantly higher than that on –7 dpi (*p* < 0.05), suggesting the importance of daily supplementation to keep a high 5-ALA concentration.

## 3. Discussion

The antiviral activity of 5-ALA was enhanced by the combined use of the PPIX exporter inhibitor FTC, indicating that the intracellular accumulation of the 5-ALA metabolite PPIX plays a major role on the antiviral activity on 5-ALA supplementation. Meanwhile, only post-treatment with 5-ALA and SFC inhibited virus growth significantly, whereas 5-ALA alone suppressed virus growth both pre-treatment and post-treatment. This difference was considered to be derived from 5-ALA metabolism. When 5-ALA is supplemented with SFC in advance, it should be reasonable that most compounds are metabolized into heme via PPIX, whereas, without SFC, they remain as PPIX instead of heme due to the lack of ferrous ions. Therefore, in the pre-treatment group, the combination of 5-ALA and SFC did not show antiviral effects, implying that PPIX is efficiently metabolized into heme under the supplementation of SFC and heme does not have antiviral effects against CSFV. Unfortunately, the antiviral effects of heme could not be directly evaluated due to its strong cytotoxicity (data not shown). In contrast, in post-treatment, PPIX is produced even in 5-ALA and SFC supplementation in the process of metabolism. This produced PPIX present not only inside but also outside the cells via ABCG transporter-mediated export might have exerted an inhibitory effect. This presumption supports the inhibitory effects of 5-ALA supplementation alone in pre-treatment and post-treatment because, in these cases, most of the supplemented 5-ALA should be just converted into PPIX in both treatment conditions and should exhibit antiviral effects. Furthermore, the antiviral effects of PPIX were observed in post-treatment, corresponding to previous studies that reported the inhibitory effects of porphyrin derivatives against several viruses belonging to the *Flaviviridae* family [11,12,13]. Taken together, PPIX is a major factor in the antiviral effects of 5-ALA against CSFV.

In previous studies, PPIX accumulation in cells by photoactivation was thought to be responsible for the antiviral effects against human immunodeficiency virus, herpes simplex virus, and vesicular stomatitis virus [14,15]. Thus, to investigate whether intracellular PPIX plays a key role in the observed antiviral effects, the production of PPIX inside and outside the cell after 5-ALA supplementation to cells was measured. The accumulation of intracellular PPIX after 5-ALA supplementation was lower than that of extracellular PPIX at 3 days post-treatment. Some clear-cell carcinoma cell lines are known to accumulate PPIX in the cell because peptide transporter 1, an uptake transporter of 5-ALA, was overexpressed in these cells. In contrast, other cell lines do not accumulate PPIX due to the high expression of ABCG transporters [16]. Moreover, in the presence of FTC, the ABCG inhibitor, the intracellular PPIX concentration increased rapidly, suggesting that SK-L cells have ABCG transporters and PPIX is released outside the cell rather than remaining inside. Low intracellular PPIX concentrations after ≤2.5 mM 5-ALA supplementation with FTC might be explained by another type of cellular transport mechanism, such as dynamin 2-mediated exocytosis in tumor cells [17]. In this study, the extracellular PPIX concentration was higher than the intracellular one; this further motivated the authors to investigate the antiviral effects of PPIX in the extracellular condition. As a result, the infectivity titer of CSFV decreased dramatically in a dose-dependent manner after 1 h incubation with PPIX at different concentrations, showing antiviral effects of extracellular PPIX. Previous studies reported that porphyrins directly affected viral particles, inhibition of virus adsorption or penetration, suppression of viral protein in other flaviviruses [11,13], and virus entry of enveloped viruses such as SARS-CoV-2 and influenza A virus [12]. The potent antiviral activity of PPIX against SARS-CoV-2 was reported via binding to the viral receptor, angiotensin-converting enzyme 2, and interfering with the interaction between viral proteins [18]. In another study, the antiviral effects of PPIX may be a result of the interaction of PPIX with the lipid bilayer of the virus envelope [12]. In CSFV, its potential viral receptor is different from SARS-CoV-2. In addition, pre-treatment and simultaneous treatment with PPIX did not suppress the virus growth of CSFV in this study; therefore, PPIX would not inhibit the virus-receptor binding but directly interact with virus particles itself and lead to the inactivation of virus particles outside the cell. As reported previously for other enveloped viruses [12], PPIX may biophysically interact with the lipid bilayer of CSFV and inhibit the membrane-associated functions required for virion-to-cell fusion. Further studies to elucidate these mechanisms are essential.

Moreover, the suppression of luciferase activities of rGPE^–^2GL by PPIX supplementation demonstrated an inhibitory effect of this compound against the viral replication cycle. PPIX is considered one of the ligands of the guanine quadruplex (G4) structures found in RNA or DNA [19]. The ligands potentially targeted G4 structures and exerted antiviral activity by interfering with viral replication and translation [20]. CSFV vALD-A76 was predicted to have 76 G4 structures in the viral genome from N^pro^ protein to 3′-untranslated region (Appendix A) according to the QGRS Mapper result; hence, similar to the previous report, the suppression of replication cycle can be attributed to the intracellular interaction with PPIX and G4s. Collectively, PPIX exhibited antiviral effects against CSFV, and its mechanism was expected as an extracellular inactivation of virus particles and intracellular inhibition of virus replication (Figure 7).

As a result of in vivo experiments, the clinical scores in High Dose were significantly lower than those of the other two groups. In addition, the period of high fever was significantly shortened in pigs of High Dose. These results are the first report that 5-ALA was able to alleviate clinical signs in animal experiments using pigs against infectious diseases. In this study, pigs were fed with 5-ALA and SFC because 5-ALA is normally used with SFC as a supplement for humans or animals, with the expectation of activating innate immunity, as mentioned above, or enhancement of health conditions [21,22]. This alleviation of clinical signs might be due to the anti-inflammatory effects of HO-1 induced after 5-ALA supplementation, which was previously confirmed by in vitro settings [23,24]. On the other hand, virus recovery from challenged pigs was almost the same among the three groups, although in vitro experiments indicated that the addition of 5-ALA and SFC at 0 dpi only once during the test had a significant inhibitory effect on CSFV replication (Figure 1C). In this animal experiments, pigs were fed with 5-ALA and SFC daily before and after virus inoculation, with the expectation of continuous supply of 5-ALA and its metabolites (e.g., PPIX and HO-1). The dose of 5-ALA in the High Dose was 10 mg/kg, which is given to humans and animals with the expectation of activating innate immunity and enhancement of health conditions. In this study, the 5-ALA levels in blood were ~10^3.5^ µg/L in pigs of High Dose (Appendix A), which might be sufficient for alleviation of clinical signs, but not to prevent pigs from CSF infection because the levels were equal to 0.1 mM and did not reach those of in vitro settings in this study. To improve the in vivo efficacy of 5-ALA against CSFV, not only supplementation dose but also supplemental duration, or supplementation of 5-ALA alone or its metabolite such as PPIX, should be further considered. Additionally, the mechanism of the alleviation of clinical signs in pigs should be evaluated using immunological and histopathological methodologies in addition to the assessment of the inhibition of virus growth.

In conclusion, this study demonstrated that both intracellular and extracellular PPIXs produced as metabolites of 5-ALA play central roles in the in vitro antiviral effects against CSFV, although the inhibitory effects of 5-ALA on viral replication were not sufficient in vivo. In general, each pig farm has individual problems caused by different pathogens. Considering that porphyrin derivatives, including PPIX, show antiviral activities against various pathogens, it is expected that 5-ALA would exert a broad spectrum of antiviral activity; therefore, 5-ALA and its metabolite would be applied as a supportive tool using better administration methods and doses for disease prevention in pigs, in combination with animal husbandry, vaccination, and biosecurity in farms.

## 4. Materials and Methods

### 4.1. Cells and Viruses

SK-L cells [25] were propagated with Eagle’s minimum essential medium (EMEM; Nissui Pharmaceutical, Tokyo, Japan) supplemented with sodium bicarbonate (Nacalai Tesque), 0.295% tryptose phosphate broth (Becton, Dickinson and Company, CA, USA), 10 mM N,N-bis-(2-hydroxyethyl)-2-aminoethanesulfonic acid (Sigma-Aldrich, St. Louis, MO, USA), and 10% horse serum (HS; Thermo Fisher Scientific, Waltham, MA, USA). This cell line was kindly provided from National Institute of Animal Health (Tsukuba, Ibaraki, Japan).

A recombinant clone of CSFV vALD-A76, derived from a virulent strain, ALD-A76, was used in this study. The vALD-A76 stain was previously established in our laboratory [26].

### 4.2. Compounds

5-ALA and SFC were provided by Neopharma Japan Co., Ltd. (Tokyo, Japan) and dissolved in EMEM containing 10% HS (10% HS EMEM) for cell culture as stock concentrations of 10 mM 5-ALA and 2.5 mM SFC. SFC provides ferrous ions essential for 5-ALA metabolism into heme caused by ferrochelatase. FTC (Catalog No. F9054, Sigma-Aldrich), an inhibitor of ABCG transporter, was dissolved in dimethyl sulfoxide (DMSO; FUJIFILM Wako Pure Chemical Corp., Osaka, Japan) at a concentration of 10 µM. PPIX (Catalog No. P562-9) was obtained from Funakoshi (Tokyo, Japan) and dissolved in DMSO as stocks at the concentrations of 10 mM and further diluted with the cell culture medium in the experimental condition. Stock solutions of all compounds were stored at −20 °C in the dark.

### 4.3. Virus Titration

SK-L cells were inoculated with the serially 10-fold diluted CSFV in 96-well plates and incubated at 37 °C for 72 to 96 h. Then, the plates were immunostained as described previously [27]. Briefly, the plate was air-dried and heat-fixed at 80 °C for 1 h. Cells were then incubated at room temperature for 1 h in the presence of the primary monoclonal antibody for NS3, 46/1 [28]. After washing with PBS, cells were incubated at 37 °C for 1 h in the presence of goat anti-mouse IgG (H + L) horseradish peroxidase conjugate (Bio-Rad, Hercules, CA, USA). The primary and secondary antibodies were diluted 2000-fold with PBS containing 1% Bovine Serum Albumin Fraction V (Roche, Basel, Switzerland) and 0.05% Tween 20 (FUJIFILM Wako Pure Chemical Corp.). Finally, cells were washed again with PBS and then stained with 3-amino-9-ethyl carbazole (Sigma-Aldrich). Virus titers were calculated and expressed as TCID_50_ per milliliter [29].

### 4.4. Virus Growth Kinetics

The growth kinetics of the recombinant vALD-A76 with or without the compounds was determined using confluent cell monolayers of SK-L cells at an MOI of 0.1. Cells were treated with 1 mM 5-ALA alone, 1 mM 5-ALA with 0.25 mM SFC or 1 µM FTC, or 100 µM PPIX during pre-treatment, simultaneous treatment, or post-treatment.

In the pre-treatment group, the compounds were added to the medium; after 24 h incubation, cells were washed with PBS, and the virus diluted by 10% HS EMEM was inoculated. After 1 h incubation with the virus, cells were washed with PBS, and a fresh medium was added. In the simultaneous treatment group, cells were washed with PBS, and the virus was diluted with 10% HS EMEM containing the compounds and added to cells. After 1 h incubation with the virus, cells were washed with PBS, and a fresh medium was added. In the post-treatment group, cells were washed with PBS, and the virus diluted by 10% HS EMEM was inoculated. After 1 h incubation with the virus, cells were washed with PBS, and a fresh medium was added. Then, the compounds were further supplemented to the medium.

Cells were incubated at 37 °C in the presence of 5% CO_2_, and their supernatants were collected at 0, 1, 2, and 3 dpi to measure the virus titers.

### 4.5. Measurement of PPIX Concentration

The PPIX concentration was measured as described previously [16]. First, 100 µL of cell suspension (5 × 10^4^ cells) were seeded into 96-well plates and cultured at 37 °C in the presence of 5% CO_2_ for 24 h. The culture media was replaced with fresh media containing 0, 1.25, 2.5, or 5 mM 5-ALA and the cells were incubated at 37 °C in the presence of 5% CO_2_. At 0 to 3 days post-treatment, the intracellular and extracellular PPIX concentrations were measured. The intracellular PPIX concentration was measured in cells treated with 100 µL of 1% sodium dodecyl sulfate after three times wash with PBS. The PPIX concentration in the cell supernatant was measured as the extracellular PPIX concentration. The fluorescence of each sample in a 100 µL was measured by POWERSCAN4 (DS Pharma Biomedical, Osaka, Japan) with emission wavelengths of 632 and 635 nm, after an excitation wavelength of 405 nm, to evaluate the intracellular and extracellular PPIX concentrations, respectively.

### 4.6. Direct Inactivation of CSFV by PPIX

The extracellular PPIX concentration against virus inactivation was investigated based on a previous study [11] with additional filtration steps. Briefly, 25 µL of vALD-A76 containing 10^6.7^ TCID_50_ was mixed with an equal volume of 100, 50, 25, 5, 1, or 0 µM PPIX and then incubated at 37 °C for 1 h. After incubation, the compound was removed from the virus diluent using illustra MicroSpin S-400 HR Columns (Thermo Fisher Scientific). The existence of PPIX in the diluent before and after filtration was confirmed by measuring fluorescence intensity as described above to confirm the removal of the compounds. The virus titer in each mixture was measured as described previously.

### 4.7. Luciferase Activity of Viral RNA Replicase Complex

The mono-cistronic replicons carrying the *Gaussia* luciferase reporter gene, pGPE^–^2GL and pGPE^–^2GL/GAA [30], were used in this study. pGPE^–^2GL/GAA is the replication-deficient replicon by impairing the active center of RNA-dependent RNA polymerase in NS5B of CSFV. Replicon RNA was transcribed in vitro from linearized plasmid DNA with *Srf*I and purified by phenol–chloroform extraction and ethanol precipitation. The purified product was used for run-off transcription using a MEGAscript^TM^ T7 kit (Thermo Fisher Scientific). After DNase I digestion and purification with illustra MicroSpin S-400 HR Columns, RNA was transfected to SK-L cells using TransIT^®^-mRNA Transfection Reagent (Mirus Bio LLC, WI, USA) according to the manufacturer’s protocol, but with modification on the RNA amount. Briefly, 2.0 × 10^5^ SK-L cells were seeded in 12-well plates and incubated overnight at 37 °C in the presence of 5% CO_2_. After incubation, 0.1 or 1.0 µg RNA was mixed with EMEM without HS, and then reagents were added as a vehicle. Cells were further incubated with or without PPIX supplemented soon after transfection at 37 °C in the presence of 5% CO_2_. Cell supernatants were collected at 3, 24, 48, and 72 h post-transfection. *Gaussia* luciferase activities were measured using the Dual-Luciferase Reporter Assay System (Promega, Madison, WI, USA) and the GloMAX Discovery System (Promega).

### 4.8. Animal Experiments

To evaluate the efficacy of 5-ALA and SFC supplementation against vALD-A76 in pigs, three pigs of High Dose, Low Dose, and control were prepared. Pigs in High Dose and Low Dose were fed daily with 5-ALA and SFC at 10 and 15.6 mg/kg and 1 and 1.56 mg/kg, respectively, from 7 days before the virus inoculation to 13 dpi. On the first and eighth days of supplementation, blood was collected from all pigs, and the 5-ALA concentration in plasma was measured based on a previous method [31]. All pigs were challenged intranasally with 10^7.0^ TCID_50_ of vALD-A76. Body temperature and clinical symptoms were monitored daily. The average and SD of body temperature of each pig before virus infection (–7 to 0 dpi) were calculated. The development of a high fever was defined as the temperature exceeding more than thrice the SD from their average temperature. Clinical signs were observed from the start to the end of the experiment according to a defined scoring system as described previously [32]. The score was confirmed by multiple researchers to minimize bias. Pigs with >20 of the clinical score would be euthanized.

At 0, 3, 5, 7, 9, 11, and 14 dpi, the blood was collected in tubes containing EDTA (Terumo, Tokyo, Japan). Nasal swabs were also collected at 0, 3, 5, 7, 9, 11, and 14 dpi. Serum was collected before the experiments and on the day of euthanasia. All pigs were euthanized at 14 dpi, except for the one that died at 13 dpi. The tissues from brains, tonsils, spleens, adrenal glands, kidneys, mesenteric lymph nodes, and colons were collected aseptically to prepare 10% homogenates in 10% HS EMEM for virus titration. The virus titers were expressed as TCID_50_ per milliliter (blood and oral swab) or per gram (tissue).

### 4.9. Statistical Analysis

To assess the statistical significance (*p* < 0.05), Tukey’s test was performed for comparison among the three groups and the Mann–Whitney U test was performed for qualitative data analysis.

### 4.10. Ethics Statement

The animal experiment was authorized by the Institutional Animal Care and Use Committee of the Faculty of Veterinary Medicine, Hokkaido University (approval no. 18-0038, approved on 26 March 2018), and performed according to the guidelines of this committee. The facilities where the animal experiment was conducted are certified by the Association for Assessment and Accreditation of Laboratory Animal Care International.

## Figures and Tables

**Figure 1 pathogens-11-00164-f001:**
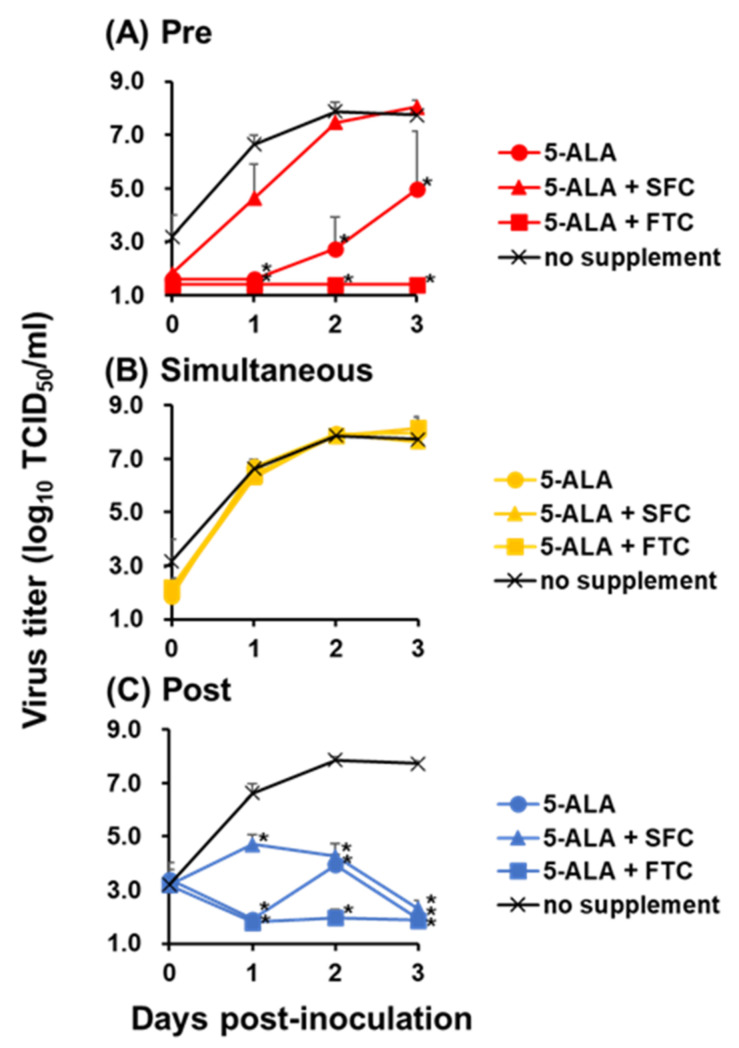
Growth kinetics of CSFV vALD-A76 with supplementation of the compounds. SK-L cells were inoculated with vALD-A76 at an MOI of 0.1 with three compounds in pre-treatment (**A**), simultaneous (**B**), or post-treatment (**C**) conditions. The final concentration of each compound was 1 mM of 5-ALA, 0.25 mM of SFC, and 1 µM of FTC. Virus titer was calculated by immunostaining at 0, 1, 2, and 3 days post-inoculation (dpi). Each line represents the mean of triplicates, with error bars indicating standard deviations (SDs). Statistical differences between nonsupplemented and supplemented cells were calculated with Tukey’s test. * *p* < 0.05, virus titer compared to nontreated cells.

**Figure 2 pathogens-11-00164-f002:**
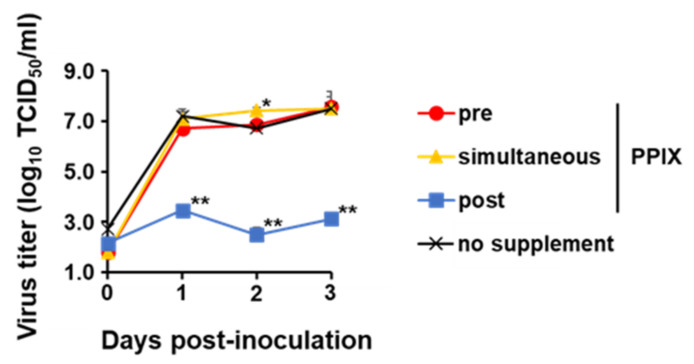
Growth kinetics of CSFV vALD-A76 with supplementation of PPIX. SK-L cells were inoculated with vALD-A76 at an MOI of 0.1 with PPIX in pre-treatment, simultaneous, or post-treatment conditions. The final concentration of PPIX was 100 µM. Virus titer was calculated by immunostaining at 0, 1, 2, and 3 dpi. Each line represents the mean of triplicates, with error bars indicating SDs. Statistical differences between nonsupplemented and supplemented cells were calculated with Tukey’s test. * *p* < 0.05; ** *p* < 0.01, virus titer compared to nontreated cells.

**Figure 3 pathogens-11-00164-f003:**
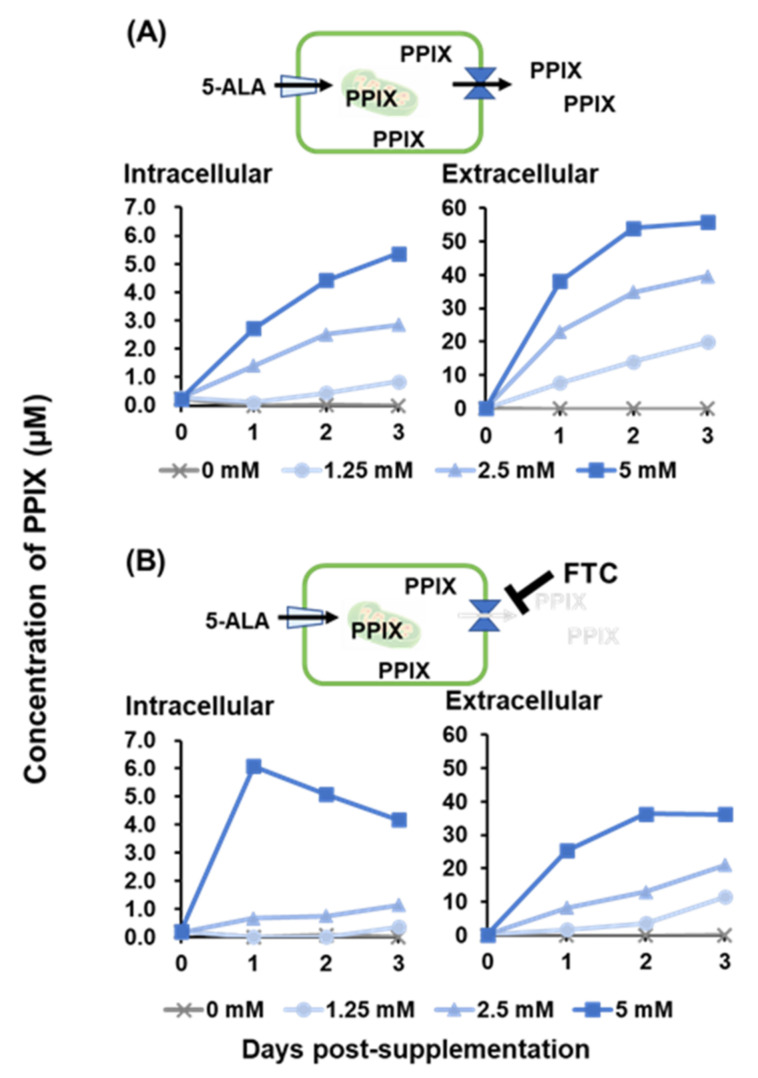
Intracellular and extracellular PPIX concentrations after the supplementation of 5-ALA with or without FTC. After supplementation of 0, 1.25, 2.5, and 5 mM 5-ALA alone (**A**) or 5-ALA with 5 µM FTC (**B**), the intracellular and extracellular PPIX concentrations at 0–3 dps were measured.

**Figure 4 pathogens-11-00164-f004:**
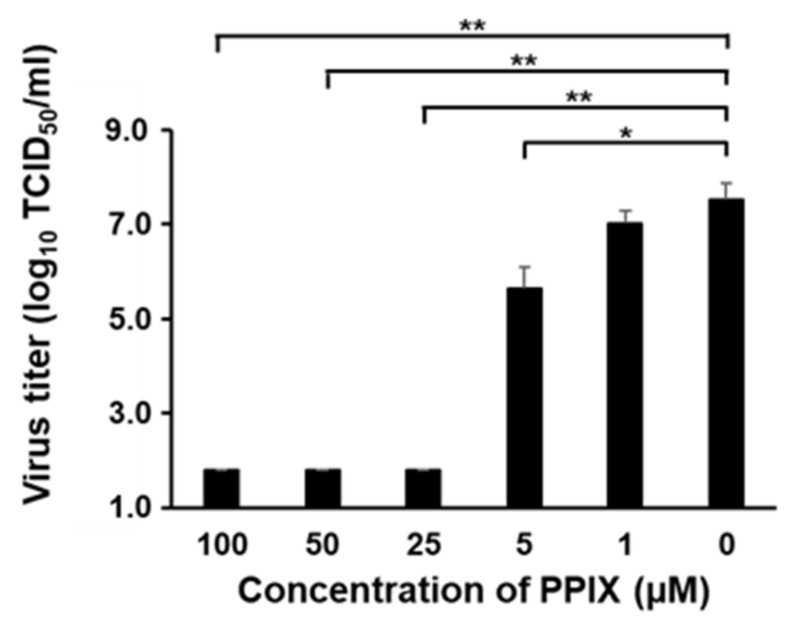
Direct inactivation effects of PPIX against CSFV vALD-A76. Twenty-five microliters of CSFV vALD-A76 containing 10^6.7^ TCID_50_ was incubated at 37 °C for 1 h with an equal volume of 100, 50, 25, 5, 1, or 0 µM PPIX. The virus titer of each mixture was measured in SK-L cells. Statistical differences between 0 µM PPIX and other concentrations of PPIX were calculated with Tukey’s test. * *p* < 0.05; ** *p* < 0.01, virus titer compared to 0 µM PPIX supplementation.

**Figure 5 pathogens-11-00164-f005:**
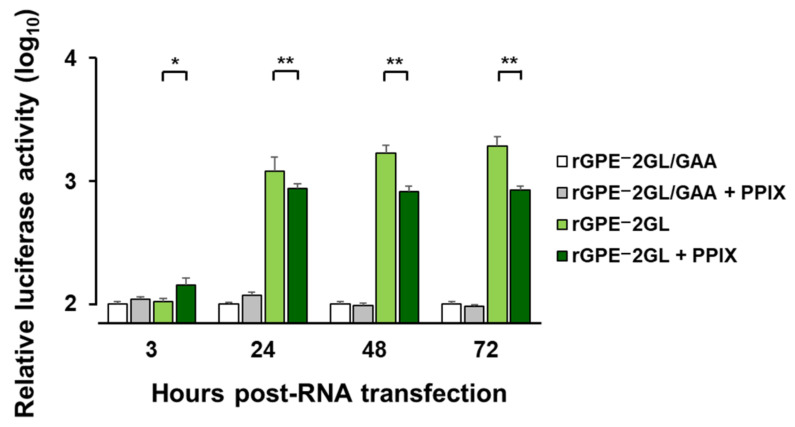
Evaluation of the inhibitory effects of PPIX on viral genome replication using the replicon-based assay. SK-L cells (2.0 × 10^5^) were transfected with 0.1 µg RNA of mono-cistronic replicon rGPE^–^2GL or replication-deficient replicon rGPE^–^2GL/GAA. PPIX was supplemented to cells soon after transfection with rGPE^–^2GL or rGPE^–^2GL/GAA at the final concentration of 100 µM, and then cells were incubated with or without PPIX. After 3, 24, 48, and 72 h incubation, the *Gaussia* luciferase activity in the supernatant, expressed as a relative luciferase activity, was measured to observe the replication cycle indirectly. The *Gaussia* luciferase activity was normalized with rGPE^–^2GL/GAA without PPIX at each time point. Each bar represents the mean of triplicates, with error bars indicating SDs. Statistical differences between the relative luciferase activity of rGPE^–^2GL without and with PPIX were calculated with Tukey’s test. * *p* < 0.05; ** *p* < 0.01, significance of luciferase activity.

**Figure 6 pathogens-11-00164-f006:**
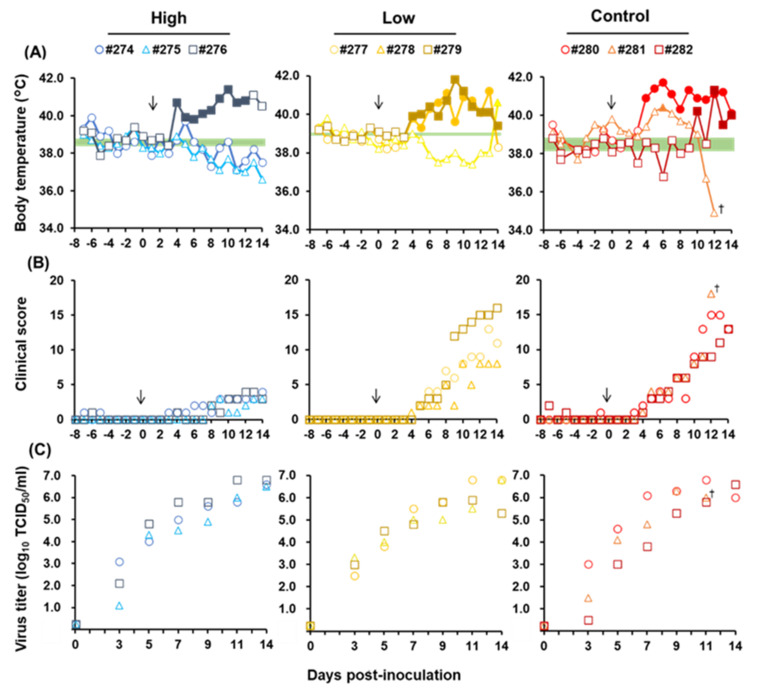
Body temperature, clinical score, and virus titer of the blood of pigs inoculated with vALD-A76. Nine 2-week-old pigs were divided into High Dose (blue), Low Dose (yellow), and Control (red). Body temperature (**A**) and clinical score (**B**) were monitored daily, and blood was collected at 0, 1, 3, 5, 7, 9, 11, and 14 dpi for the measurement of virus titer (**C**). All pigs were euthanized at 14 dpi, except for one (#281). The average and SD of the body temperature of each pig were calculated from −7 to 0 dpi. High fever was defined as the body temperature exceeding the average with thrice the SD after virus infection and is shown in the filled marker. Clinical signs were scored from 10 parameters, with a range of 0–3 for each. Virus titer in blood was measured in SK-L cells. The black arrow indicates the date of virus inoculation. The green bar in (A) represents the average temperature of pigs in the group. ^†^ Pig (#281) died at 13 dpi.

**Figure 7 pathogens-11-00164-f007:**
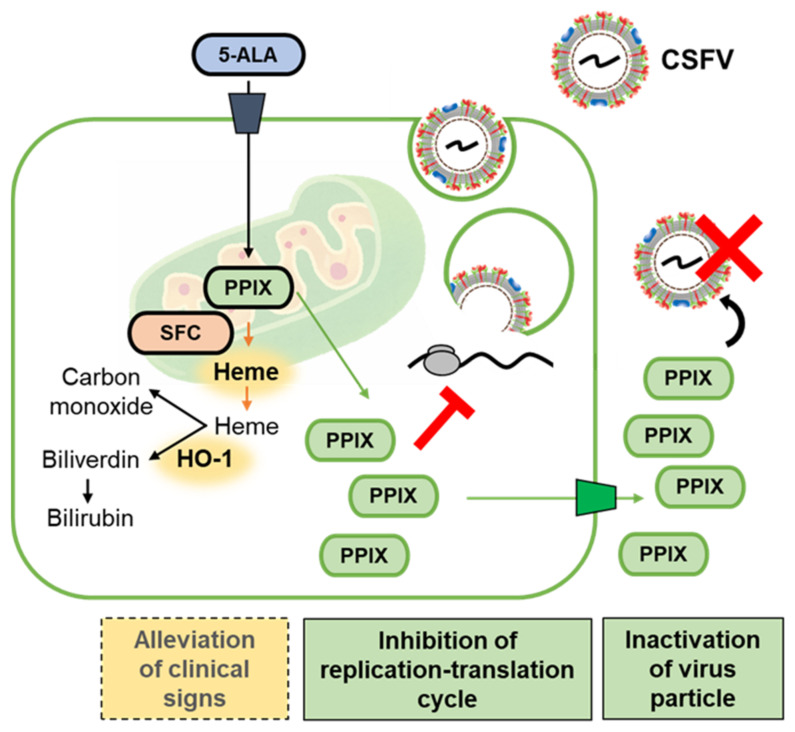
Graphical summary of the antiviral and possible palliative effects of 5-ALA. When 5-ALA is supplemented externally, 5-ALA is mainly converted to PPIX, and this compound is secreted to the cytosol and outside the cells. PPIX showed in vitro antiviral effects by inactivating virus particles extracellularly and inhibiting the viral replication cycle intracellularly. With supplementation of 5-ALA with SFC, 5-ALA is metabolized mainly to heme, and HO-1 is induced, which would work for the alleviation of clinical signs in pigs.

## Data Availability

Data are contained within the article or Appendix A.

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
