# Peer review of "Antiviral Effects of 5-Aminolevulinic Acid Phosphate against Classical Swine Fever Virus: In Vitro and In Vivo Evaluation"

_pathogens, 2022, doi:10.3390/pathogens11020164_

Round 1

Reviewer 1 Report

The manuscript by Hirose et al. focuses on the assessment of the in vitro and in vivo antiviral efficacy of of 5-ALA against CSFV. It highlights the inhibitory effect of PPIX, the metabolite of 5-ALA, on CSFV replication. However, some of the materials and methods are not appropriate described. Also, some of the results are not clearly presented. Here are some points:

Major concern:

  1. Explain how do authors determine the concentration of each compound used in the treatments.
  2. Explain why do authors use different concentrations of FTC in figure 1 (1Um) and figure 3 (5uM)?
  3. Line 280, do authors mean that no PPIX is produced in the pretreatment with 5-ALA and SFC? Why? If PPIX is produced in cells pretreated with 5-ALA and SFC, why no inhibition of virus growth is observed? Did authors test the PPIX concentration under the conditions of pre- and post-treatment with 5-ALA and SFC?
  4. Line 316-320, explain the possible mechanisms of “lead to the inactivation of virus particles”.
  5. Provide the source of cells and viruses, cat log# of the reagents used in this study, especially the compounds.
  6. Line 388-396, provide more details of the experiment, like the concentration of primary and secondary antibody, dilution buffers for these antibodies, blocking buffer, etc..
  7. Line 423-431, provide more details of the experiment, including the cell number, amount of the cell supernatant or cell lysate used for PPIX concentration test.

Minor concern:

  1. Line 25, change “Virus” to “CSFV”
  2. Line 27, the description “The infectious titer….with PPIX” is unclear. Please rewrite it.
  3. Line 45, change “had” to “have”
  4. Line 69, explain “naturally synthesized amino acid”
  5. Line 115, change “cells” to “SK-L cells”
  6. Line 119-120, change “even…” to “through the time course of experiment”
  7. Line 145, change “in” to “of”
  8. Line 152, change “concentration” to “concentration of PPIX”
  9. Line 157, change “point of action” to “action point”
  10. Line 185, change “than” to “than that”
  11. Line 198, indicate the amount of virus used for incubation.
  12. Line 237, unclear description, rewrite it.
  13. Line 244, change “of” to “in”
  14. Line 270, change “in” to “on the”, change “on” to “of”
  15. Line 295, move “at 3 days posttreatment” to the behind of “extracellular PPIX”
  16. Line 298, change “others” to “other cell lines”
  17. In supplemental document, for the EC50, change “against each strain of” to “of each compound against CSFV”

Author Response

Dear Reviewer 1,

Thank you for your kind suggestion. The corrections to the major comments are highlighted in light blue. Corrections to minor comments are highlighted in yellow.

<Major comments>

  1. Explain how do authors determine the concentration of each compound used in the treatments.

Answer: Thank you for your comment. Based on the evaluation of EC50 against CSFV and other previous studies, the final concentrations of 5-ALA, SFC and FTC were determined to be 1 mM, 0.25 mM, and 1 µM, respectively.

Modification: As you pointed out, additional explanation has been added to lines 112-114 and 133.

  1. Explain why do authors use different concentrations of FTC in figure 1 (1 µM) and figure 3 (5 µM)?

Answer: Thank you for your comment. In the study of Figure 1, 1 µM of FTC was used in the experiment along with 1 mM of 5-ALA. In the study of Figure 3, implemented concentration of FTC was raised to 5 µM since the maximum amount of 5-ALA was 5 mM. The authors realized that this information needs to be added to the text.

Modification: As you pointed out, additional explanation has been added to lines 164 and 174-176.

  1. Line 280, do authors mean that no PPIX is produced in the pretreatment with 5-ALA and SFC? Why? If PPIX is produced in cells pretreated with 5-ALA and SFC, why no inhibition of virus growth is observed? Did authors test the PPIX concentration under the conditions of pre- and post-treatment with 5-ALA and SFC?

Answer: Thank you for your comment. We understood that we could not explain clearly the metabolism of 5-ALA under the supplementation of SFC. When 5-ALA is supplemented with SFC in advance, most compounds are metabolized into heme via PPIX, whereas, without SFC, they remain to be PPIX instead of heme due to the lack of ferrous ions. Therefore, in the pretreatment group, the combination of 5-ALA and SFC did not show antiviral effects, implying that PPIX is efficiently metabolized into heme under the supplementation of SFC. To minimize misunderstanding as the reviewer suggested, we added one sentence in this discussion although we have no data about the PPIX concentration under the conditions of pre- and post-treatment with 5-ALA and SFC

Modification: As you pointed out, an additional discussion has been added to lines 284 and 286-288.

  1. Line 316-320, explain the possible mechanisms of “lead to the inactivation of virus particles”.

Answer: I agree with your comment. We have added our discussion as the reviewers pointed out.

Modification: As you pointed out, an additional discussion has been added to lines 330-333.

  1. Provide the source of cells and viruses, cat log# of the reagents used in this study, especially the compounds.

Answer: I agree with your comment. We have added the information as the reviewers pointed out.

Modification: As you pointed out, additional information has been added to lines 392-394, 396-397, 402-403, and 404-405.

  1. Line 388-396, provide more details of the experiment, like the concentration of primary and secondary antibodies, dilution buffers for these antibodies, blocking buffer, etc..

Answer: I agree with your comment. We have added the information as the reviewers pointed out.

Modification: As you pointed out, additional information has been added to lines 416-418.

  1. Line 423-431, provide more details of the experiment, including the cell number, amount of the cell supernatant or cell lysate used for PPIX concentration test.

Answer: We understood that this comment was related to the measurement of the concentration of PPIX in paragraph 4.5, although the reviewer indicated Line 423-431(paragraph 4.6). We have added the information as the reviewers pointed out in paragraph 4.5.

Modification: As you pointed out, additional information has been added to lines 439-444 and 447.

<Minor comments>

  1. Line 25, change “Virus” to “CSFV”
  2. Line 27, the description “The infectious titer….with PPIX” is unclear. Please rewrite it.
  3. Line 45, change “had” to “have”
  4. Line 69, explain “naturally synthesized amino acid”
  5. Line 115, change “cells” to “SK-L cells”
  6. Line 119-120, change “even…” to “through the time course of experiment”
  7. Line 145, change “in” to “of”
  8. Line 152, change “concentration” to “concentration of PPIX”
  9. Line 157, change “point of action” to “action point”
  10. Line 185, change “than” to “than that”
  11. Line 198, indicate the amount of virus used for incubation.
  12. Line 237, unclear description, rewrite it.
  13. Line 244, change “of” to “in”
  14. Line 270, change “in” to “on the”, change “on” to “of”
  15. Line 295, move “at 3 days posttreatment” to the behind of “extracellular PPIX”
  16. Line 298, change “others” to “other cell lines”
  17. In supplemental document, for the EC50, change “against each strain of” to “of each compound against CSFV”

Answer: Thank you for your comments. All the points given by all the reviewers have been corrected according to the instructions.

Modification: All modifications of minor comments were highlighted in yellow.

Reviewer 2 Report

Hirose and colleagues studied the effect of 5-aminolevulinic acid phosphate on the replication of classical swine fever (CSF) virus. The authors used in vitro model and then they evaluated the antiviral activity of this compunds in pigs. Although the results ot in vitro studies are promosing, the in vivo results are not as expected. The viral load and body temperatues were comparable in the groups of pigs asminisrtated or/not the supplements in differenet doses.

The manuscript has a good flow and well written. However, major points need to be addressed

1- In vitro studies: Figure 1(C): It is not clear when the drug was added post. The protocol in methodology is not clear at this point.

Also, since the compound gives some effect Pre and/or post but not simultaneous, these findings need to be addressed in the discussion section, Why???

2-Page 6 lines 190-194: "As a result, the virus titer of CSFV was suppressed in a dose-dependent manner from 5 μM PPIX treatment, and finally, it was under the detection limit at a dose of >25 μM (Figure 4),"

This part is not accurate according to Fig4: there is not data on dose response curve from PPIX starting from 5uM to 25 uM. I guess these data need to be included.

3- My major point is the in vivo study: a) What are the parameters of clinical scores that the authors calculate? please include in details. 

b) I guess the in vivo studies either need to be deleted or repeated oe expanded to include other effects since there are no effect on the compound in vivo 

c) Alternatively, the authors can assess the viral load in diiferent organs collected and see if there is effect in the viral load inside organs. 

d) OR if there are some histology slides processed from the experimental pigs that can show the degree of inflammation or clinical signs will be helpful

Author Response

Dear Reviewer 2,

Thank you for your kind suggestion. The corrections to the major comments are highlighted in light green.

<Major comments>

1-(1) In vitro studies: Figure 1(C): It is not clear when the drug was added post. The protocol in methodology is not clear at this point.

Answer: I agree with your comment. We have added the information as the reviewers pointed out.

Modification: To clarify the methodology, additional information has been added to lines 427-435.

1-(2). In vitro studies: Figure 1: Also, since the compound gives some effect Pre and/or post but not simultaneous, these findings need to be addressed in the discussion section, Why???

Answer: This study revealed that the substance that inhibits the growth of CSFV is neither 5-ALA nor SFC, but PPIX, a metabolite of 5-ALA. Even if 5-ALA or SFC is added at the same time, PPIX is not present in the culture medium or in the cells under this condition. This could be the reason why no antiviral effect was observed.

Modification: According to the suggestion, the explanation for this group was added into lines 130-133.

  1. Page 6 lines 190-194: "As a result, the virus titer of CSFV was suppressed in a dose-dependent manner from 5 μM PPIX treatment, and finally, it was under the detection limit at a dose of >25 μM (Figure 4),"

This part is not accurate according to Fig4: there is not data on dose response curve from PPIX starting from 5uM to 25 uM. I guess these data need to be included.

Answer: Thank you for your suggestion. It is true that there is not enough data in Figure 4 to describe it as “a dose-dependent manner”. Considering the purpose of this experiment, we thought we should remove the word "in a dose-dependent manner " rather than adding the new data between 5μM and 25μM.

Modification: According to the suggestion, we have removed "in a dose-dependent manner" at lines 197-199.

3-1. My major point is the in vivo study: a) What are the parameters of clinical scores that the authors calculate? please include in detail.

Answer: The scoring method reported by Mittelholzer et al. (Reference no. 32) is widely used as an objective and animal welfare-friendly method for evaluating the pathogenicity of CSFV in pigs. We conducted scoring in accordance with this method. Therefore, we believe that there is no need to repeat the methodology of the scoring in this paper.

Modification: No modification.

3-2

b) I guess the in vivo studies either need to be deleted or repeated or expanded to include other effects since there are no effect on the compound in vivo

Answer: Thank you for your suggestion. As a result of in vivo experiments, the clinical scores in High Dose were significantly lower than those of the other two groups. In addition, the period of high fever was shortened in pigs of High Dose. These results are the first report that 5-ALA was able to alleviate clinical signs in animal experiments using pigs against infectious diseases, although the virus recovery from challenged pigs was almost the same among the three groups.

The previous descriptions in this paragraph did not explain the novelty of this animal experiment clearly. We have added several sentences to this paragraph along with one new reference. In addition, we discussed the points that need to be improved in order to demonstrate the antiviral effect against CSFV based on the results of this animal experiment.

Modification: We did not delete in vivo study but explained the novel point of this study and future challenge for the inhibition of CSFV growth at lines 354-355, 362-364, 366-369, and 374-376. A new reference, Ito et al. (2018) was added as reference No. 22.

c) Alternatively, the authors can assess the viral load in different organs collected and see if there is effect in the viral load inside organs.

Answer: Additional studies were conducted to recover viruses from oral swabs and organ homogenates. As a result, there was no significant decrease in virus recovery for the High Dose or Low Dose groups. Although these additional experiments did not provide results to suggest inhibition of viral replication, the animal experiments were worthwhile because a significant reduction in clinical signs was observed, as described above and below.

Modification: The information of additional studies was described at lines 254-255, 494-499, and 514-515. Figure S3 was added for the virus recovery from oral swabs and organ homogenates.

d) OR if there are some histology slides processed from the experimental pigs that can show the degree of inflammation or clinical signs will be helpful

Answer: Unfortunately, in our animal experiments, we did not collect organs for histopathological investigation. A histopathological investigation is necessary for further experiments.

In terms of clinical signs of pigs, the clinical scoring by the method of Mittelholzer et al. (Reference no. 32) is an established method for pathogenicity tests in pigs of CSFV. The score was confirmed by multiple researchers to minimize bias. In addition, the average and SD of body temperature of each pig before virus infection (–7 to 0 dpi) were calculated and the development of a high fever was defined as the temperature exceeding more than thrice the SD from their average temperature. We believe that these methods provide strong objective evidence that can explain the alleviation of clinical signs in pigs treated with 5-ALA.

Modification: To enhance the validity of the scoring, one sentence was added in the materials and methods at lines 490-491. The criteria for the fever in pigs has been added at lines 242-245. In addition, we also added the requirement of immunological and histopathological investigation to clarify the mechanism of the alleviation of clinical signs in pigs in future experiments at lin

Round 2

Reviewer 1 Report

The authors address all the comments in the revised manuscript. This revised version with better quality of presentation can be easily understood.

Reviewer 2 Report

The authors replied to my questions adequately.